# Quantitative evaluation of protective antibody response induced by hepatitis E vaccine in humans

Gui-Ping Wen[1,5], Linling He[2,5], Zi-Min Tang[1,5], Si-Ling Wang[1,5], Xu Zhang[1], Yuan-Zhi Chen[1], Xiaohe Lin[2], Chang Liu[1,3], Jia-Xin Chen[1], Dong Ying[1,3], Zi-Hao Chen[1], Ying-Bin Wang[1], Wen-Xin Luo[1,3], Shou-Jie Huang[1], Shao-Wei Li [1,3], Jun Zhang[1], Zi-Zheng Zheng [1✉], Jiang Zhu [2,4✉] & Ning-Shao Xia [1,3✉]

Efficacy evaluation through human trials is crucial for advancing a vaccine candidate to clinics. Next-generation sequencing (NGS) can be used to quantify B cell repertoire response and trace antibody lineages during vaccination. Here, we demonstrate this application with a case study of Hecolin®, the licensed vaccine for hepatitis E virus (HEV). Four subjects are administered the vaccine following a standard three-dose schedule. Vaccine-induced anti-bodies exhibit a high degree of clonal diversity, recognize five conformational antigenic sites of the genotype 1 HEV p239 antigen, and cross-react with other genotypes. Unbiased repertoire sequencing is performed for seven time points over six months of vaccination, with maturation pathways characterize for a set of vaccine-induced antibodies. In addition to dynamic repertoire profiles, NGS analysis reveals differential patterns of HEV-specific anti-body lineages and highlights the necessity of the long vaccine boost. Together, our study presents a quantitative strategy for vaccine evaluation in small-scale human studies.

[1] State Key Laboratory of Molecular Vaccinology and Molecular Diagnostics, National Institute of Diagnostics and Vaccine Development in Infectious Diseases, School of Public Health, Xiamen University, 361005 Xiamen, Fujian, PR China. [2] Department of Integrative Structural and Computational Biology, The Scripps Research Institute, La Jolla, CA 92037, USA. [3] School of Life Sciences, Xiamen University, 361005 Xiamen, Fujian, PR China. [4] Department of Immunology and Microbiology, The Scripps Research Institute, La Jolla, CA 92037, USA. [5] These authors contributed equally: Gui-Ping Wen, Linling He, Zi-Min Tang, Si-Ling Wang. ✉email: zhengzizheng@xmu.edu.cn; jiang@scripps.edu; nsxia@xmu.edu.cn

Vaccines are one of the greatest achievements in medicine and have eradicated many infectious diseases. However, empirical vaccine strategies have yet to overcome global health challenges posed by influenza, human immunodeficiency virus type-1 (HIV-1), malaria, and tuberculosis, among others[1]. Even for licensed vaccines, optimal dosage, schedule, and antigen combination remain issues of considerable debate[2,3]. Various strategies have been explored to reduce the "shot burden", to simplify the vaccine schedule, and to develop combination vaccines that can provide protection against multiple diseases with fewer shots[2,3]. In HIV-1 vaccine research, a rational strategy has contributed to the recent advances[4,5]. As one of the key cornerstones in this strategy, next-generation sequencing (NGS) has provided insights into the diversity and evolution of broadly neutralizing antibodies (bNAbs) during both infection[6,7] and vaccination[8–11]. NGS analysis of antibody response during vaccination may facilitate the investigation of the mechanism of vaccine protection[1,12,13] and optimization of the vaccine regimen, including antigen, adjuvant, route of inoculation, and prime-boost strategies.

Hecolin®, the only licensed vaccine against hepatitis E virus (HEV) with an efficacy of ~100% in the phase III trial with >100,000 participants[14], provides a template for examining the utility of an antibody-based strategy for vaccine evaluation. Transmitted by the fecal-oral route and blood transfusion[15], HEV can lead to acute and chronic hepatitis E and is associated with many extrahepatic manifestations[16]. HEV-related illness may cause severe or fatal disease in pregnant women with a mortality of up to 25% and high rates of spontaneous abortion and still-birth[16]. Among the four major HEV genotypes[15], *1* and *2* circulate exclusively in humans and cause large outbreaks in resource-limited countries, while *3* and *4* are zoonotic and cause sporadic cases of autochthonous acute hepatitis E. Despite the notable differences in epidemiological characteristics and hosts, the four HEV genotypes belong to a single serotype[15]. The genome of HEV contains three open-reading frames (ORFs), of which ORF2 encodes a major viral capsid protein of 660 amino acids (aa) associated with virion assembly, host interaction, and immunogenicity[15]. The E2s domain (aa 459–606) of the capsid protein forms a homodimer on the virus surface, mediates virus–host interactions, and presents a major target for immune recognition[17–19]. All E2s domains adopt a similar structure[20,21], with an aa-level sequence identity of 83–97% across different genotypes. A truncated form (aa 368–606) of the capsid protein from *genotype-1* HEV, which contains the E2s domain and is termed p239(1), is the antigen in Hecolin®[14]—the vaccine that protects against HEV *genotypes 1* and *4* in a phase III trial[14,22]. The safety study of Hecolin® in the healthy US adult population has been approved by the Food and Drug Administration (FDA) (Identifier: NCT03827395). In recent studies, different immunogenicity profiles were reported for proteins encoded by *genotype-1* and *4* ORF2s[23]. Serological analysis of vaccinated humans has been performed to assess anti-HEV immunoglobulin G (IgG) titer, anti-HEV IgG avidity, and epitope specificity[14,18,22]. However, it remains to be determined whether the vaccine derived from p239(1) can protect against all four genotypes with the same efficacy. Furthermore, the neutralizing activity, epitope specificity, protective mechanism, and temporal development of HEV vaccine-induced antibody response in humans has not been fully characterized at the molecular level.

Here, we conduct a small-scale clinical study to probe the human antibody response to HEV vaccination in a systematic manner. We administer four human subjects three doses of Hecolin® and perform detailed serological analysis. We utilize a quantitative strategy to characterize the vaccine-induced antibody response by combining antigen-specific single-cell sorting and antibody cloning with NGS-based antibody-repertoire profiling and lineage-tracing analysis. Our study has provided valuable insights into the HEV vaccine mechanism and will have important implications for future vaccine evaluation in humans.

## Results

**HEV-specific serum response in human vaccination**. Four individuals aged 19–25 years with no prior history of HEV infection or vaccination were given three doses of Hecolin® at months 0, 1, and 6 and were followed-up for a period of 376 days after the first dose (Fig. 1a). Overall, vaccination induced a vigorous anti-HEV IgG response in all four donors. Positive anti-HEV IgG seroconversion was observed at day 30 after the first dose, with anti-HEV IgG levels ranging from 0.17 World Health Organization (WHO) unit per mL (WU per mL) to 0.89 WU per mL (Fig. 1b). Anti-HEV IgG levels peaked at day 221 or 247 (4.62–11.71 WU per mL) and gradually decreased to ≥ 1.48 WU per mL as measured at day 376 after the first dose. Consistently, western blot showed that anti-HEV antibodies were enriched in all four donors during the vaccination, with a clear preference for binding to the p239(1) dimer (Supplementary Fig. 1). For each donor, the anti-HEV IgG avidity increased from 7.52–16.73% at day 30 to 53.07–70.07% at day 221 (Fig. 1c), suggesting a continuous maturation process due to multiple vaccine doses. The plasmablast response to HEV vaccination was assessed by fluorescence-activated cell sorter (FACS) (Supplementary Fig. 2). Overall, plasmablasts expanded rapidly within the first 7 days after each dose but decreased significantly by 1 month (Supplementary Fig. 3), as they became antibody-producing plasma cells (PCs) or memory B cells that would give rise to the next plasmablast response[24]. The percentage of plasmablasts versus total peripheral B cells ranged from 0.55% to 0.83% at 1 week after the third dose. All four donors showed increasing serum neutralizing activity over time from an undetectable level before vaccination to the 50% inhibitory dose ($ID_{50}$) values of 281–1367 at day 221 after the first dose (Fig. 1d). Taken together, the results indicated that the HEV vaccine, Hecolin®, can induce a robust immune response with rapid development of p239(1)-specific antibodies in all four donors.

**Single-cell isolation of HEV p239(1)-specific antibodies**. To assess the memory B-cell response to HEV vaccination, peripheral blood mononuclear cells (PBMCs) at 1 month after the third dose were stained with the fluorescently labeled vaccine antigen, p239(1), and analyzed by FACS (Supplementary Fig. 2). The results indicated that 0.21–0.31% of memory B cells (defined as CD3−/CD20+/CD27+) were IgG+ and p239(1)-specific at 1 month after the third dose (Supplementary Fig. 4). A total of 104 p239(1)-specific monoclonal antibodies (mAbs), 21–32 from each donor, were obtained from these B cells using single-cell polymerase chain reaction (PCR) (Supplementary Tables 1 and 2). In this study, the mAb name was defined as the donor index (A, B, C, and D for donors 1, 2, 3, and 4, respectively) followed by the clone number, for example, A103 (Supplementary Table 2). In terms of V gene usage (Fig. 2a; Supplementary Fig. 5), *IGHV1-69* appeared to be predominant, ranging from 21.43% to 43.48% per donor. *IGHV1-18* and *IGHV3-30* accounted for 4.76–17.86% and 3.57–9.52% per donor, respectively. *IGHV4-59* represented 9.38% and 14.29% of the mAbs isolated from donors 1 and 4, respectively. Of note, most of the p239(1)-specific mAbs (52.38–81.25%) from donors 1, 2, and 4 used κ-light chains (Supplementary Fig. 6). The degree of somatic hypermutation (SHM) ranged from 10.3 to 21.0 nucleotide substitutions per $V_H$ gene (Fig. 2b), similar to anti-RSV and anti-influenza mAbs[25,26]. The heavy-chain complementarity-determining region 3 (CDRH3) length ranged from 9 to 25 aa (Fig. 2c),

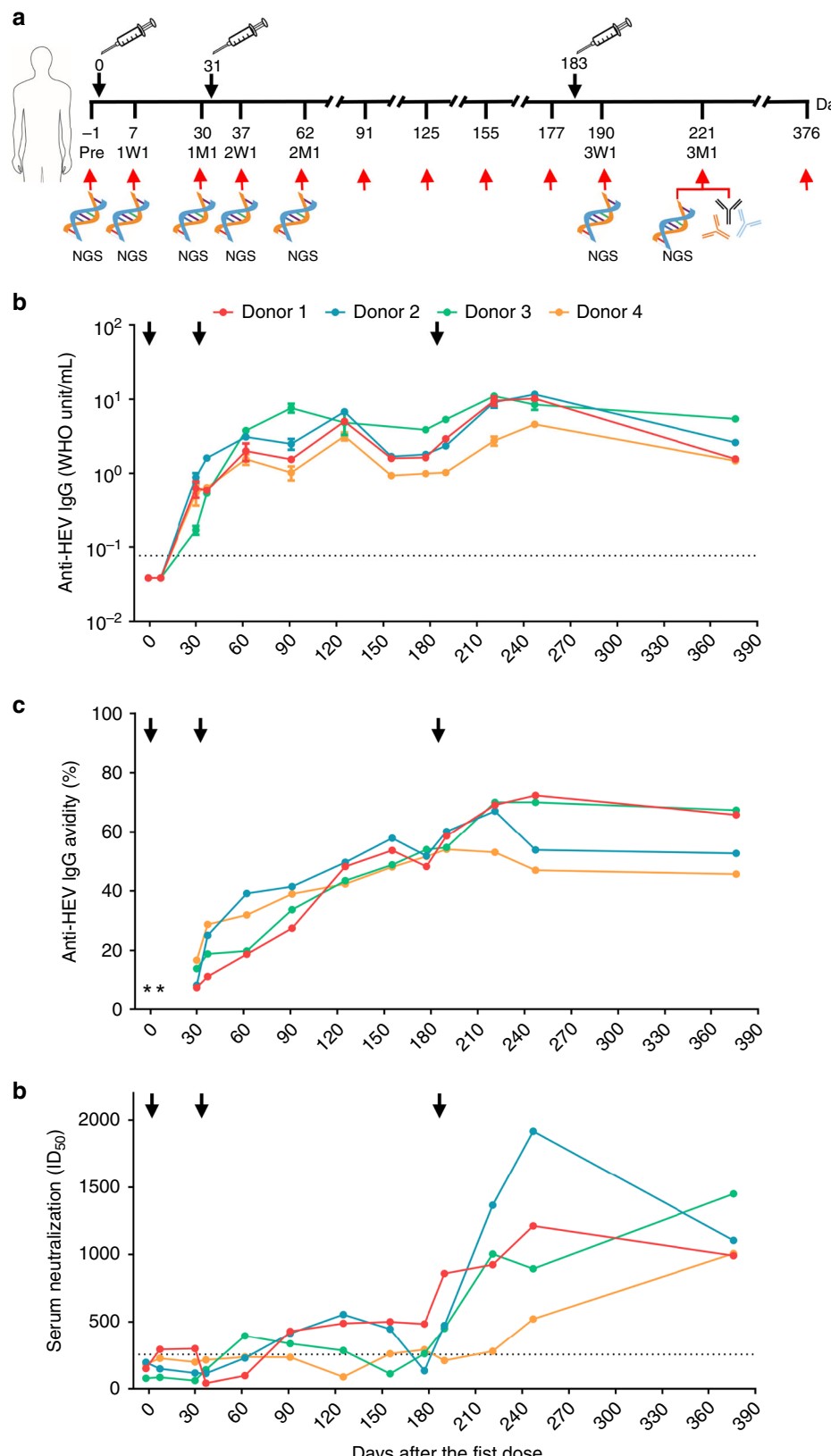

in comparison with an average of 13.5 aa observed for healthy human donors[7]. IgG1 accounted for >39.1% of the mAbs from each donor, with IgG2 and IgG4 being the other two major sub-types (Supplementary Fig. 5). Irrespective of the original IgG subtype, all mAbs were expressed as full-length IgG1s for in vitro functional validation, such as antigen binding and virus neu-tralization. To summarize, the HEV vaccine Hecolin® elicited p239 (1)-specific mAbs of diverse IgG subtypes with preferential germ-line gene usage, relatively low levels of SHM, and canonical CDRH3 length.

**Fig. 1 Study design and clinical measures of HEV vaccine response. a** Immunization regimen and critical time points in the longitudinal analysis of human B-cell response to the HEV vaccine, Hecolin®. The first, second, and third doses are indicated above the time axis at days 0, 31, and 183, respectively, with black arrows and needles. The seven time points selected for antibody-repertoire sequencing are indicated below the time axis with red arrows and "NGS." Vaccine-induced, p239(1)-specific monoclonal antibodies (mAbs) were isolated from samples collected 1 month after the third dose (3M1), which is labeled below the time axis. Dynamics of anti-HEV IgG titer (**b**), anti-HEV IgG avidity (**c**), and serum neutralization measured by the 50% inhibitory dose (ID$_{50}$) (**d**) during vaccination. Error bars indicate standard deviation (s.d.) in (**b**). The dash lines represent the detection limit for the anti-HEV IgG and the threshold of neutralizing capacity for the serum neutralization assays. *Only anti-HEV IgG positive sera were tested for anti-HEV IgG avidity. Source data are provided in the Source Data file.

**Vaccine-induced mAbs show broad and potent neutralization.** We first characterized the binding activity of each mAb against the p239 protein of *genotypes 1*, *3*, and *4* by enzyme-linked immunosorbent assay (ELISA). Overall, most mAbs demonstrated medium-to-strong binding to p239(1) (EC$_{50}$ ≤ 100 ng per mL), with 23.8–65.6% showing high reactivity (EC$_{50}$ ≤ 10 ng per mL) (Fig. 2d). Since the p239 protein from *genotype 2* cannot assemble into virus-like particles (VLPs), the recombinant E2 proteins were used in ELISA. All four genotypes could be recognized by these mAbs, despite notable variations in binding affinity (Supplementary Fig. 7). Nonetheless, our results suggest that these broadly reactive mAbs in the donor repertoires may contribute to the cross-genotype protection conferred by Hecolin®. More than 66% of the mAbs from each donor exhibited detectable HEV neutralization (IC$_{50}$ < 200 µg per mL), with 8.7–50.0% designated as high-potency mAbs (IC$_{50}$ ≤ 0.5 µg per mL) (Fig. 2e). Effective neutralization of *genotype 3* HEV suggests that Hecolin® will likely protect against this genotype during natural infection. Further analysis revealed a significant correlation between p239 binding and HEV neutralization (for p239(1): $P < 0.0001$, $r = 0.409$; for p239(4): $P < 0.0001$, $r = 0.424$) (Fig. 2f), but not between SHM and HEV neutralization, suggesting that extensive maturation is not required for the vaccine-induced antibody response to achieve effective virus neutralization (Supplementary Fig. 8). The p239(1)-specific mAbs from donors 1 and 3, with moderate SHM rate and CDRH3 length, showed higher binding and neutralizing activities than those from donors 2 and 4, who exhibited opposite antibody patterns. Specifically, while mAbs from donor 4 showed the highest SHM rate and the shortest CDRH3 length among the four donors, mAbs from donor 2 appeared to have the lowest SHM rate and the longest CDRH3 loops. The analysis of mAb sequences highlights the importance of proper SHM rate and CDRH3 length to the function of HEV vaccine-induced mAbs. Collectively, our functional analyses demonstrated that Hecolin® can effectively induce antibodies with broad and potent neutralizing activity due to their ability to recognize and bind the capsid protein of different HEV genotypes.

**Main target for cross-genotype HEV p239(1)-specific mAbs.** Ten antigenic sites on the p239(1) protein have been identified[18], with four linear epitopes in the region of aa 403–457[18] and six conformational epitopes in the protruding E2s domain. Here, all p239(1)-specific mAbs bound the E2s domain that contains the receptor-binding site (RBS) (Supplementary Fig. 9), confirming that E2s is the target for vaccine-induced antibody response[27,28]. Since E2s is conserved across genotypes[20], this finding may explain the vaccine-conferred protection against diseases caused by both *genotypes 1* and *4* and support the notion of a single HEV serotype. In addition, all mAbs showed stronger binding to dimeric p239(1) than monomeric p239(1), consistent with the donor serum analysis (Supplementary Fig. 10). This was in line with the previous reports that cross-genotype NAbs recognize conformational epitopes on the dimeric interface of E2s[17,18,21].

The dimerization of the E2s domain was also found to be essential for the virus–host interaction and disease progression[17].

We performed a detailed epitope mapping of the p239(1)-specific mAbs as previously described[18]. Each mAb was assigned to a specific antigenic site based on the combined results from antibody competition and alanine scanning. Overall, the p239(1)-specific mAbs targeted five antigenic sites (C1, C2, C3, C5, and C6) (Fig. 3a, b). A large proportion of mAbs, 46.4–91.3% from each donor, recognized antigenic site C2, an immunodominant epitope of the HEV p239(1) protein. Based on alanine scanning, mAbs that recognized C2 could be divided into five subtypes, with C2–1 and C2–3 being the two most prevalent subtypes (Fig. 3c; Supplementary Fig. 11a). In addition, 4.3–14.3% of the mAbs recognized antigenic site C5, while a small fraction (up to 25.0%) targeted antigenic site C6. Sequence analysis revealed conserved sequence signatures for those mAbs directed to C2-2/3 and C6 (Supplementary Figs. 11b–d), reminiscent of the recent report of conserved antibody signatures in human vaccination[11]. In summary, our results indicate that most p239(1)-specific mAbs target antigenic sites C2, C5, and C6, which are immunodominant in HEV vaccine-induced antibody response and likely associated with specific antibody genetic features. Lastly, we analyzed the relationship between neutralizing activity and antigenic site (Fig. 3d). Overall, the neutralizing mAbs appeared to target antigenic sites C2 and C6, with the C6-directed mAbs showing the highest potency. In comparison, the C2-directed mAbs demonstrated a broad range of neutralizing activity, with 49.3% showing medium-to-high potency (IC$_{50}$: 0.001–4.294 µg per mL). Weak or non-neutralizers were found to mainly recognize antigenic sites C3 and C5.

**Analysis of B-cell repertoire response to HEV vaccination.** Long-read NGS and antibody bioinformatics have enabled in-depth analysis of B-cell responses during infection and vaccination[6–11]. Here, we investigated the B-cell response to HEV vaccination in four donors across seven time points (Fig. 1a)— 1 week and 1 month after each dose (termed 1W1, 1M1, 2W1, 2M1, 3W1, and 3M1) plus pre-vaccination (termed Pre)—using the unbiased repertoire analysis[7]. Because CDRH3 offers an unambiguous fingerprint for identifying an antibody lineage[29,30], we focused on the heavy-chain analysis and obtained 1.18–2.32 million sequences for each time point (Supplementary Table 3). Longitudinal repertoire profiles were generated to facilitate the characterization of HEV vaccine-induced B-cell response.

In terms of germline gene usage (Fig. 4a), four donors exhibited similar patterns of distribution irrespective of the time points, with 43–66% of heavy chains originating from *IGHV1*, *IGHV3*, and *IGHV4* (Supplementary Fig. 12a). These results were consistent with our previous analysis of five healthy normal donors[31], suggesting that these germline genes are involved in frequent pathogen encounters and contribute to B-cell memory. As *IGHV4* and *IGHV1* were preferably activated in the repertoires of donors 1 and 3, respectively, a broader germline gene activation was noted for donors 2 and 4 during the vaccination (Fig. 4a). A visible increase in germline gene frequency (≥40%

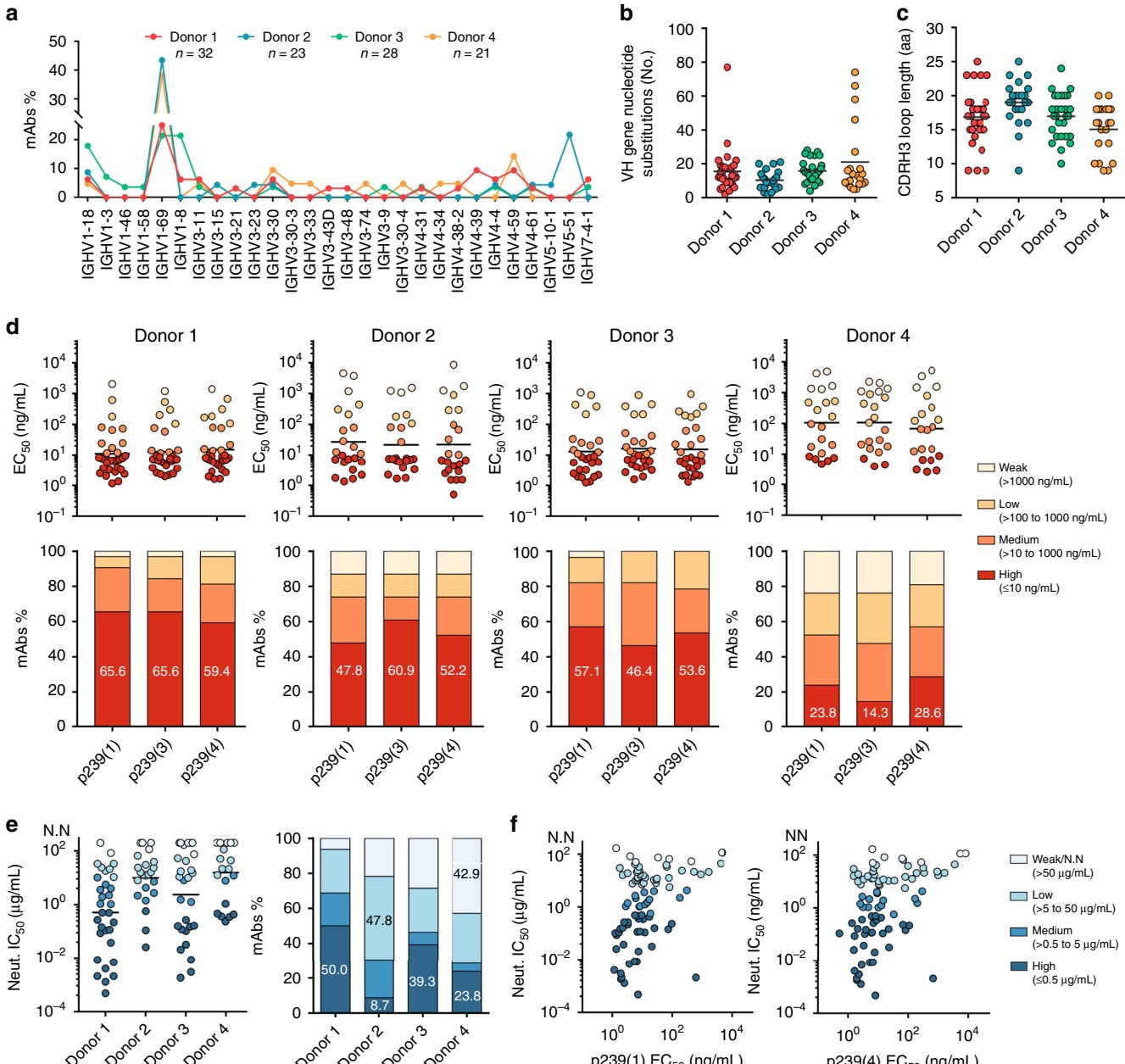

**Fig. 2 Systems analysis of HEV vaccine-induced p239(1)-specific mAbs. a** Distribution of heavy-chain variable (VH) germline genes. Donors 1, 2, 3, and 4 are indicated by red, cyan, green, and orange, respectively. **b** Distribution of nucleotide mutations in the VH gene. Black bars indicate the mean value. **c** Distribution of heavy-chain complementarity-determining region 3 (CDRH3) lengths (amino acids, aa). Black bars indicate the mean value. **d** Antibody-binding activity ($EC_{50}$) measured for the p239 proteins of three genotypes (1, 3, and 4) in four ranges (upper panel), and percentage of mAbs with the indicated $EC_{50}$ range (lower panel). Black bars indicate the geometric mean value. **e** Antibody-neutralizing activity ($IC_{50}$) in four ranges (left) and percentage of mAbs with the indicated $IC_{50}$ range (right). N.N. denotes non-neutralizing. Black bars indicate the geometric mean value. **f** Correlation of neutralizing activity ($IC_{50}$) and binding activity ($EC_{50}$) measured for p239(1) (left) and p239(4) (right). $IC_{50}$ is divided into four ranges as in (**e**). Each colored circle in (**b**) to (**f**) represents a mAb. The number of mAbs shown in (**b**) to (**e**) is 32 for donor 1, 23 for donor 2, 28 for donor 3, and 21 for donor 4, respectively. Vaccine-specific but non-neutralizing mAbs are excluded in (**f**) as their $IC_{50}$ values cannot be accurately determined. A total of 88 mAbs are shown in (**f**). Spearman rank-correlation analysis was used to analyze the relationship between p239 binding and HEV neutralization (for p239(1): $P < 0.0001$, $r = 0.409$; for p239(4): $P < 0.0001$, $r = 0.424$) in (**f**). Source data are provided in the Source Data file.

than the previous time point) often occurred at 1 week after each injection (Fig. 4a). Of note, *IGHV1-69*, which was used predominantly by the p239(1)-specific mAbs (Fig. 2a), was activated at least once during the HEV vaccination. *IGHV4-34* and *IGHV4-39* exhibited a similar pattern, consistent with their prevalence in the general B-cell repertoire. In terms of SHM or germline divergence (Fig. 4b; Supplementary Fig. 12b), all four

donors produced a large population of near-germline (SHM: 0–4%) antibodies after the first or the second dose, while showing a higher degree of SHM (on average 10.0–11.3%) after the third dose, which can be explained by the expansion and diversification of vaccine-induced B cells. In terms of CDRH3 length, antibodies with 17-aa to 19-aa CDRH3 loops were enriched in all four donors after the third dose (Fig. 4c; Supplementary Fig. 12c),

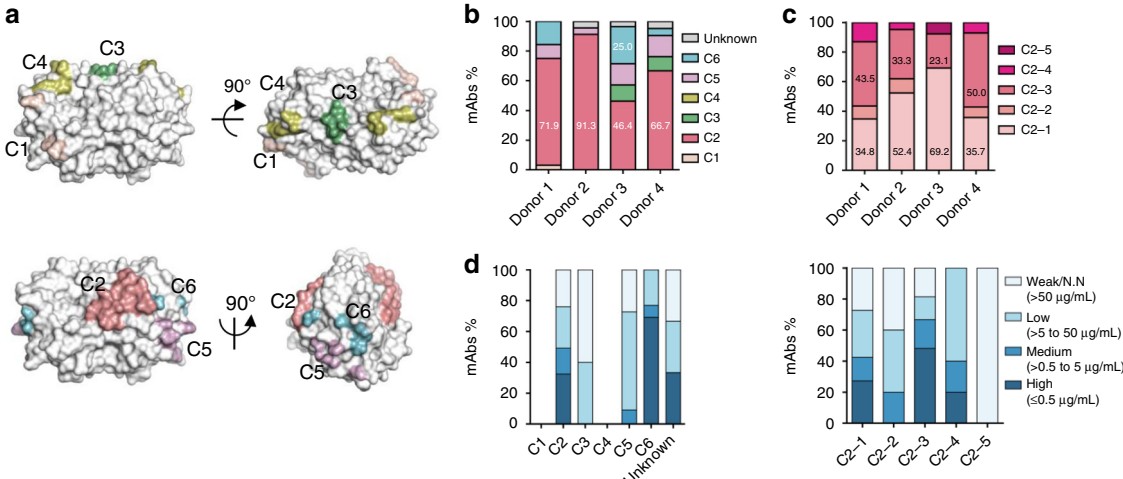

**Fig. 3 Epitope-function mapping of HEV vaccine-induced p239(1)-specific mAbs. a** Location of six distinct antigenic clusters (C1–C6) on the protruding domain of the HEV capsid. The color-coding scheme is described as follows: C1 (pink), C2 (red), C3 (green), C4 (yellow), C5 (purple), and C6 (blue). **b** Percentage of mAbs directed to each antigenic site, color-coded as in (**a**). **c** Percentage of mAbs directed to five subtypes of antigenic site C2. **d** Antibody-neutralizing activity (IC$_{50}$) plotted against the antigenic site. Left: C1–C6; right: C2–1 to C2–5. IC$_{50}$ is divided into four ranges as shown in Fig. 2e, f. The antigenic sites recognized by at least two mAbs are analyzed in (**d**). Source data are provided in the Source Data file.

consistent with the finding that the majority of p239(1)-specific mAbs isolated from 3M1, while showing a low level of SHM (Supplementary Fig. 13a), possess long CDRH3 loops (Supplementary Fig. 13b). Taken together, the NGS analysis provided a systematic, repertoire-level view of human B-cell response to HEV vaccination.

**Global analysis of antibody lineage development.** We traced the lineages of p239(1)-specific mAbs in donor repertoires across seven time points to probe the ontogeny of vaccine-induced antibody response (Fig. 5). To this end, we adopted a CDRH3-based lineage definition, with CDRH3 identity of 90% or greater and CDRH3 length variation of two residues or less. By this definition, 43 of the 104 p239(1)-specific mAbs could be traced in donor repertoires that demonstrated substantial neutralizing activity (Fig. 5a). In contrast, somatic variants could not be identified for 61 mAbs, suggesting that they may not represent a prevalent antibody response to HEV vaccination. Most of the traceable mAbs were derived from *IGHV1-69* (Fig. 5b–d; Supplementary Fig. 14). While most of the traceable mAbs from donors 1, 2, and 4 recognized antigenic site C2, 75.0% of the traceable mAbs from donor 3 were directed to antigenic site C6 (Fig. 5e). The third dose appeared to be crucial for donors 1 and 2, with 63.2% and 38.5% of the traceable mAbs first detected in their 3W1 or 3M1 repertoires (Fig. 5f), respectively. Nearly 45% of the p239(1)-specific mAb lineages peaked in size at 1 month after the first dose. More than 91.3% of the p239(1)-specific mAb lineages peaked at 1 week after the second and the third dose and returned to the baseline within a month (Fig. 5g; Supplementary Fig. 15), consistent with a typical plasmablast response[24]. Similar patterns of antibody response have been reported for influenza and HBV vaccinations[9,10]. For donor 4, no significant lineage expansion was observed for p239(1)-specific mAbs after the third dose. Consistently, the anti-HEV IgG level in donor 4 did not increase at 3W1, but instead, at 3M1 (Fig. 1a). The delay of HEV-specific antibody response in donor 4 might be caused by an unknown infection as indicated by Flu-like symptoms around the third dose, suggesting that a concurrent infection may affect the human immune response to vaccination[2]. For donors 1 and 2, 59.4% and 56.5% of the p239(1)-specific mAbs were traceable and were often present in excess in repertoires (e.g., 1314 somatic

variants in a lineage for donor 1 and 2150 for donor 2, respectively), suggesting a large circulating plasmablast population. In contrast, only 14.3% and 33.3% of the p239(1)-specific mAbs were traceable in the repertoires of donors 3 and 4, with small lineages for mAbs form donor 3, indicating a moderate plasmablast response. Taken together, our analysis revealed the complex connections between lineage traceability and size—two simple metrics of lineage prevalence within repertoire—and antibody function, such as epitope specificity and neutralizing activity in a time-dependent manner.

**HEV vaccine-induced cross-donor antibody lineage patterns.** We generated two-dimensional (2D) identity-divergence plots to visualize mAb lineages in the context of donor repertoires across seven time points, which allowed us to identify conserved lineage patterns shared by multiple donors (termed cross-donor). The most recognizable pattern is the evolving B-cell response: namely, once an mAb was identified in the donor repertoire, somatic variants could be found for this mAb at 1 week after each of the following doses. Approximately 5.3–25.0% of the traceable mAbs from donors 1, 2, and 3 were detected after the first dose and exhibited an evolving B-cell response to HEV vaccination. Using A275 as an example (Fig. 6a, panel 1; Supplementary Fig. 15), the A275 lineage was first observed at 1M1, suggesting that this lineage arose from naive B cells in response to the first dose. After the second dose (2W1), the A275 lineage dramatically expanded to form a large island extending from the main population on the 2D plot. Although the A275 lineage was below the detectable level at 2M1, it expanded again at 3W1. A larger fraction (15.4–28.6%) of the traceable mAbs were first detected after the second dose and exhibited an evolving B-cell response for donors 1, 2, and 4 (Fig. 6a, panel 2; Supplementary Fig. 16a, panel 2). Using A278 as an example, a population of 220 A278-like heavy chains was observed at 2W1 (Fig. 6a, panel 2; Supplementary Fig. 15). As this mAb lineage continued to evolve, a slight expansion was observed at 3W1 that exhibited higher degree of SHM and lower sequence identity relative to A278. Notably, the majority (25.0–63.2%) of the traceable mAbs from donors 1, 2, and 3 were only identified after the third dose (Fig. 6a, panel 3; Supplementary Fig. 16a, panel 3), representing the most common pattern of B-cell response. Since all these p239(1)-specific mAbs were detected

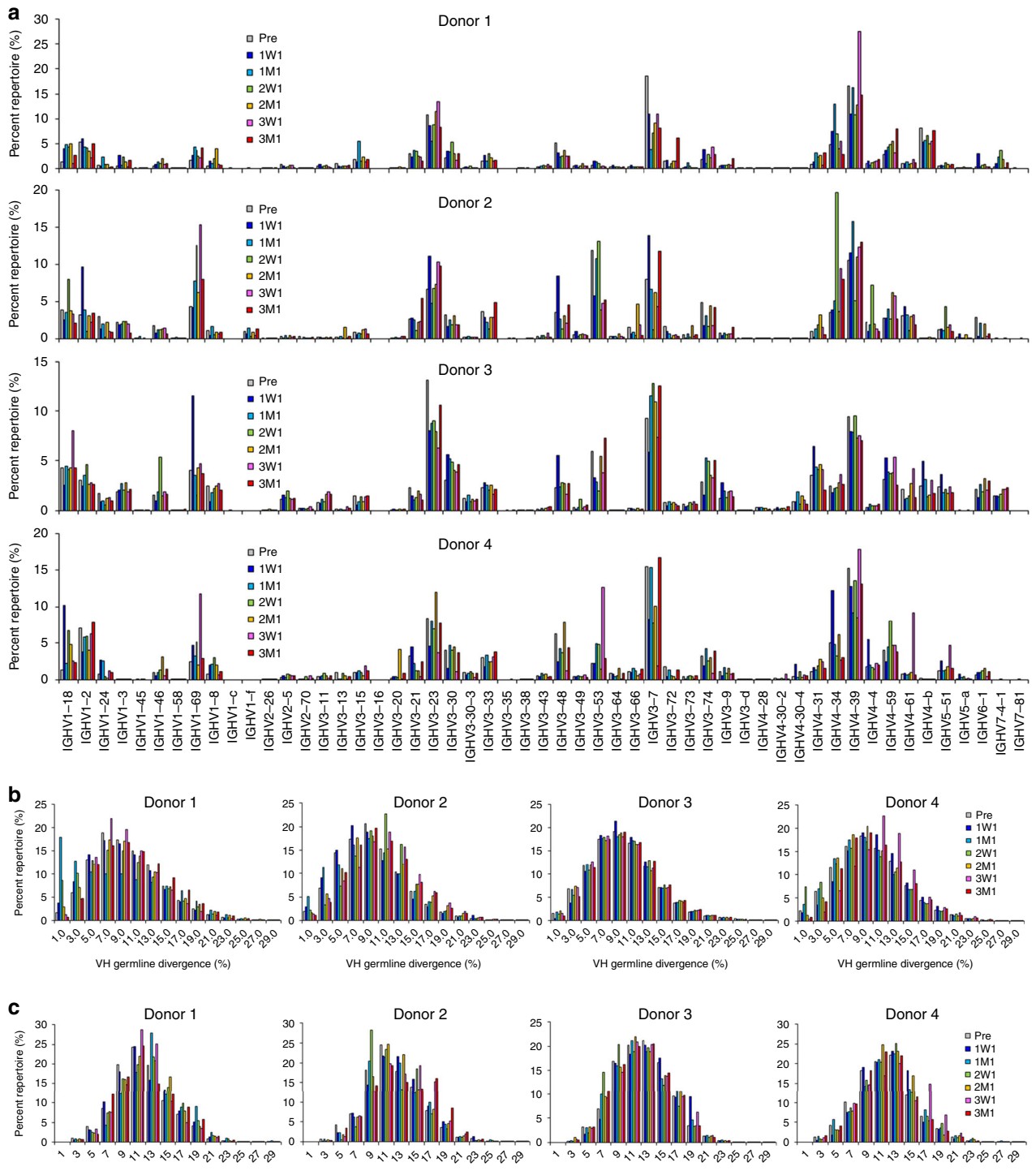

**Fig. 4 Unbiased analysis of B-cell repertoire response during the course of vaccination.** For each donor, the temporal distribution is plotted for (**a**) heavy-chain variable (VH) germline gene, (**b**) VH-germline divergence, or degree of somatic hypermutation (SHM), and (**c**) heavy-chain complementarity-determining region 3 (CDRH3) loop length. Pre: pre-vaccination; 1W1: 1 week after the first dose; 1M1: 1 month after the first dose; 2W1: 1 week after the second dose; 2M1: 1 month after the second dose; 3W1: 1 week after the third dose; 3M1: 1 month after the third dose. The color-coding scheme is described as follows: Pre (gray), 1W1 (blue), 1M1 (cyan), 2W1 (green), 2M1 (orange), 3W1 (purple), and 3M1 (red). Source data are provided in the Source Data file.

after the first vaccine dose, they were most likely induced by the HEV vaccination. Interestingly, one mAb from donor 2 (B137) and one mAb from donor 4 (C339) were found in the Pre repertoires, suggesting that they might originate from pre-existing HEV-reactive B cells (Fig. 6a, panel 4, and Supplementary Fig. 16a, panel 4). Both mAbs showed lineage expansion and

maturation throughout the vaccination, indicating that pre-existing immunity may play a role in vaccine response.

All traceable mAbs mentioned above showed persistent B-cell development and were detectable at 3W1. However, 14.3–15.4% of the traceable mAbs from donors 2 and 4 were found at 1W1 and 2W1, but not after the third dose (Fig. 6a, panel 5;

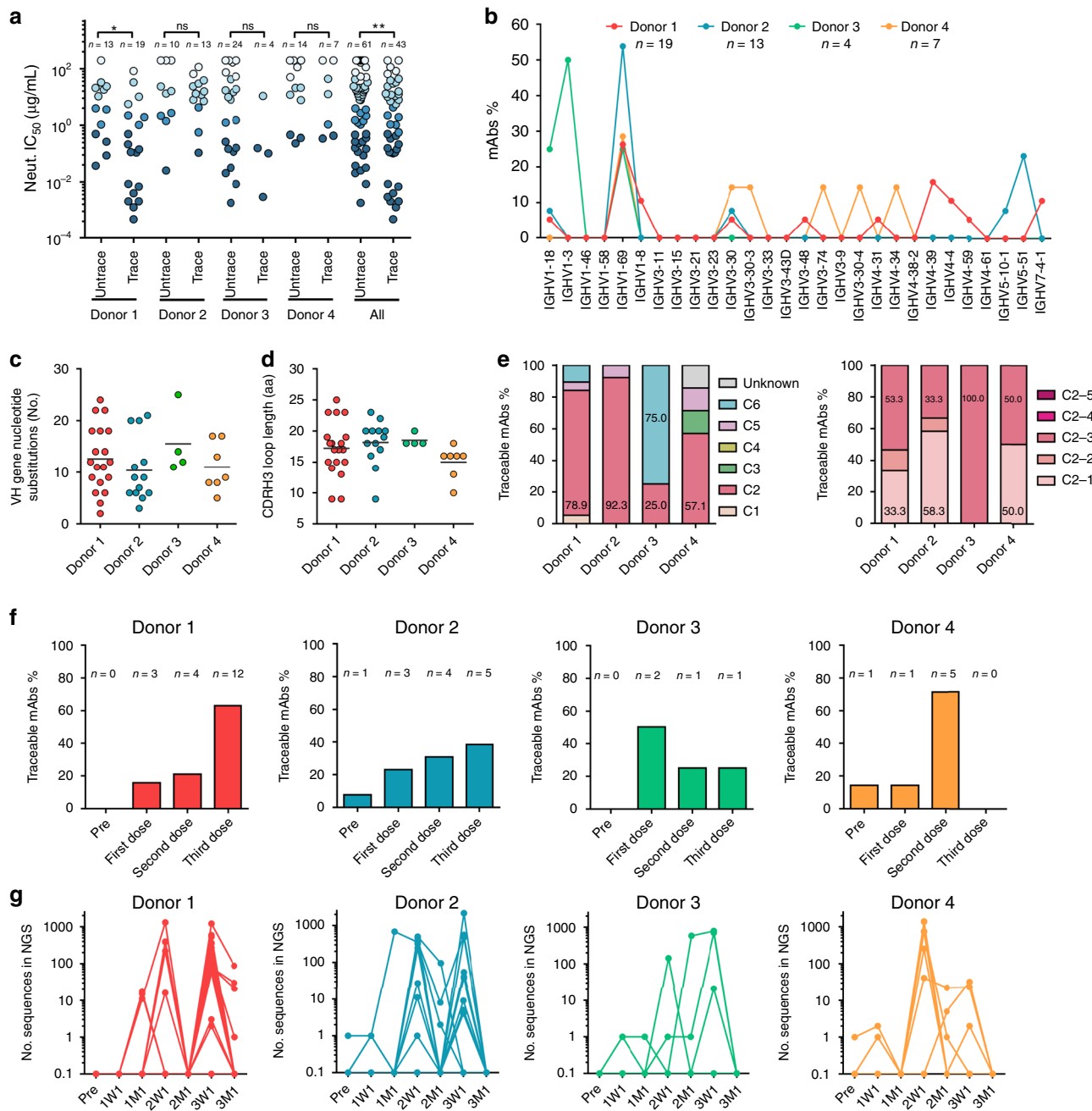

**Fig. 5 An overview of HEV p239(1)-specific mAbs in terms of lineage traceability in the unbiased repertoires. a** Analysis of neutralizing activity between lineage-traceable and untraceable mAb groups. There are 19 lineage-traceable mAbs from donor 1, 13 from donor 2, 4 from donor 3, and 7 from donor 4. There are 13 lineage-untraceable mAbs from donor 1, 10 from donor 2, 24 from donor 3, and 14 from donor 4. **b** Distribution of traceable heavy-chain variable (VH) germline genes. Donors 1, 2, 3, and 4 are indicated by red, cyan, green, and orange, respectively. **c** Distribution of nucleotide mutations in the traceable VH gene. Black bars indicate the mean value. **d** Distribution of heavy-chain complementarity-determining region 3 (CDRH3) lengths (amino acids, aa). Black bars indicate the mean value. There are 19 mAbs from donor 1, 13 from donor 2, 4 from donor 3, and 7 from donor 4 in (**c**) and (**d**) with each colored circle indicating an mAb. **e** Epitope analysis of lineage-traceable mAbs for four donors. **f** Time points at which the HEV-specific mAbs were first identified in the donor repertoires. **g** Temporal distribution of lineage size, measured as the number of CDRH3-defined somatic variants, of the traceable HEV p239(1)-specific mAbs in the donor repertoires. Pre: pre-vaccination; 1W1: 1 week after the first dose; 1M1: 1 month after the first dose; 2W1: 1 week after the second dose; 2M1: 1 month after the second dose; 3W1: 1 week after the third dose; 3M1: 1 month after the third dose. Source data are provided in the Source Data file.

Supplementary Fig. 16a). Furthermore, 15.4–42.9% of the traceable mAbs from donors 2, 3, and 4 showed a transient B-cell response and were only identified at 2W1 (Fig. 6a, panel 6; Supplementary Fig. 16a). The mAbs that were undetectable after the third dose showed lower binding to the vaccine antigen than those present in the donor repertoires after the third dose

(Supplementary Fig. 17). These results suggest that these mAb lineages became undetectable after the third dose due to their lower fitness in competition with other HEV-specific mAb lineages.

Representative mAbs were analyzed using phylogenetic tools (Fig. 6b–f; Supplementary Figs. 16b–g and Supplementary

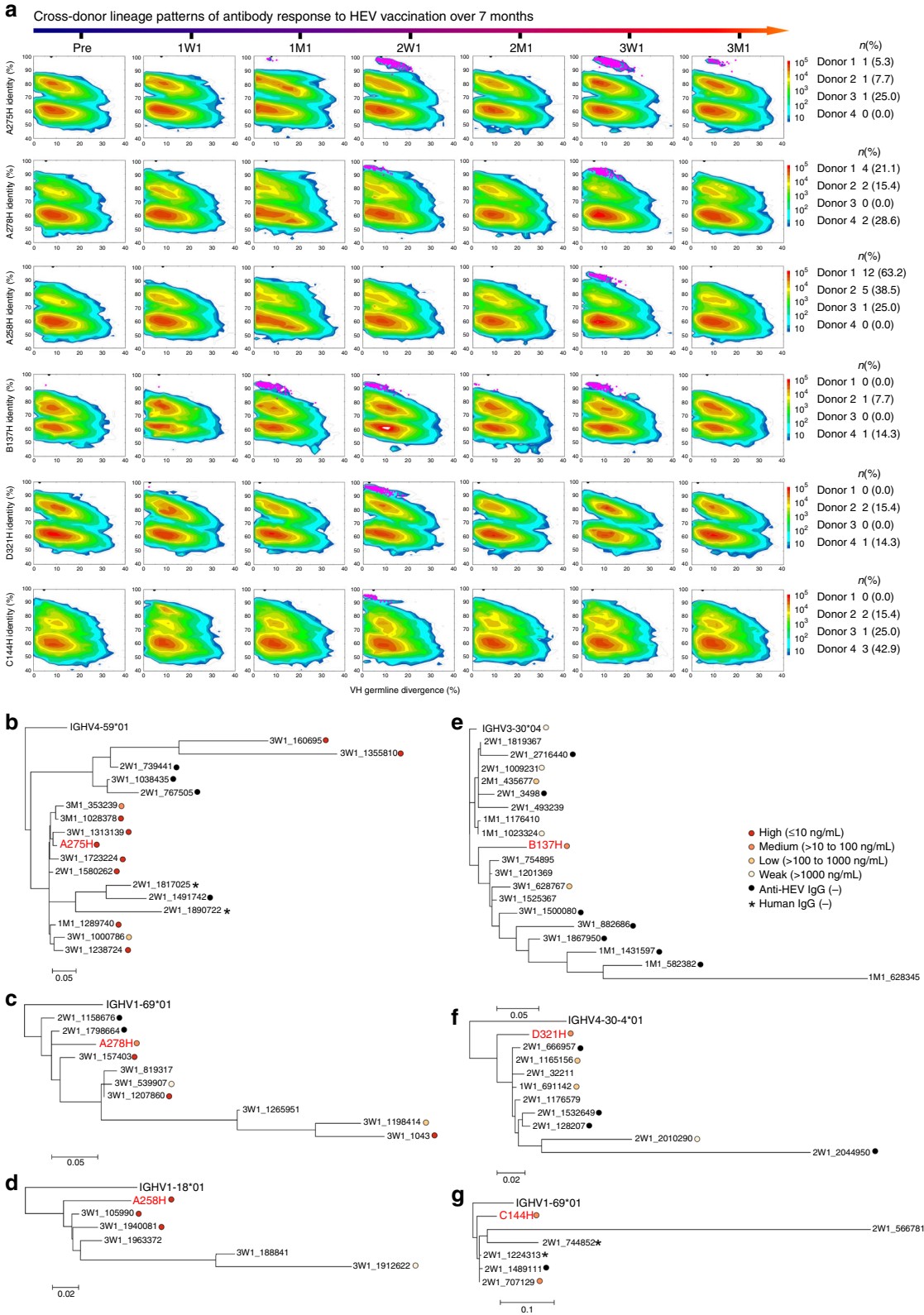

**a** Cross-donor lineage patterns of antibody response to HEV vaccination over 7 months

Tables 4–7). Dendrograms of somatic variants selected from the lineages of A275 (Fig. 6b), A170 (Supplementary Fig. 16b), A137 (Fig. 6e), and A283 (Supplementary Fig. 16c) showed that the sequences derived from the second or the third dose were clustered with the sequences derived from the previous dose, suggesting a recall of memory B cells. In the A275 dendrogram,

some sequences derived from the second and the third dose formed a separate branch (Fig. 6b), indicative of lineage diversification. The A278 dendrogram displayed a branching pattern with a "chronological" order: while the 2W1 variants were only found from the top branches, the 3W1 variants resided on those more distant branches (Fig. 6c). Selected NGS-

**Fig. 6 Cross-donor patterns of HEV p239(1)-specific antibody lineage development during vaccination. a** Identity-divergence plots of the A275, A278, A258, B137, D321, and C144 lineages in the context of unbiased donor B-cell repertoires. Heavy chains are plotted as a function of sequence identity (%) to the reference mAbs and sequence divergence (%) from their putative germline genes. Color-coding indicates sequence density on the two-dimensional (2D) plot. Somatic variants of A275, A278, A258, B137, D321, and C144 identified by a heavy-chain complementarity-determining region 3 (CDRH3) identity cutoff of 90% are shown as magenta dots on the 2D plots. Pre: pre-vaccination; 1W1: 1 week after the first dose; 1M1: 1 month after the first dose; 2W1: 1 week after the second dose; 2M1: 1 month after the second dose; 3W1: 1 week after the third dose; 3M1: 1 month after the third dose. Dendrograms of selected heavy-chain variants from the lineages of A275 (**b**), A278 (**c**), A258 (**d**), B137 (**e**), D321 (**f**), and C144 (**g**) rooted by their putative germline variable (V) genes. Reconstituted mAbs are labeled with dots if they can be expressed and bind the HEV vaccine antigen, p239(1). The parental mAb heavy chain is labeled in red on the dendrogram, which was generated using the maximum-likelihood (ML) method. (EC$_{50}$ [ng per mL]: ≤ 10, red circle; > 10 to 100, orange circle; > 100 to 1000, aurantia circle; > 1000, yellow circle; p239(1)-nonspecific, black).

derived heavy chains were paired with their parental light chains for functional validation. Overall, most reconstituted mAbs (from the early time points) showed comparable or reduced antigen binding with respect to the parental mAbs, suggesting that the last dose may be crucial for these antibody lineages to achieve high affinity. For A275 and A283, the reconstituted mAbs and the parental mAbs bound p239(1) with similar EC$_{50}$ values, suggesting that some antibody lineages can acquire high affinity rapidly and remain active during the vaccination. Overall, the results indicate that the three-dose schedule can effectively stimulate vaccine-induced memory B cells to expand and affinity mature, in addition to engaging naive B cells upon each boost.

**HEV vaccine-induced donor-specific antibody lineage patterns.** In addition to cross-donor patterns (Fig. 6), we also identified some lineage patterns specific to individual donors (Supplementary Fig. 18). Donor 1 was a "strong vaccine responder" and provided many examples as to how vaccine-induced B-cell lineages may follow diverse maturation pathways. While the A225 lineage exhibited an oscillating pattern in lineage size, the A130 lineage only expanded transiently in the 1M1 repertoire. In donor 3, the C172 lineage was detectable at 1W1 and 1M1, fell below the detectable level at 2W1, and expanded dramatically at 2M1, following a unique course of development during the vaccination. In most cases, the reconstituted mAbs showed comparable or reduced antigen binding relative to their parental mAbs, similar to those mAbs presenting the cross-donor lineage patterns. These donor-specific patterns highlight the complexity of vaccine-induced antibody response in such clinical studies that may be attributed to various factors such as genetics, sex, and health condition of the subjects.

## Discussion
In vaccine development, clinical trials need to be conducted in phases to determine the safety, immunogenicity, and efficacy of a vaccine candidate. The outcome of a clinical trial can be affected by many factors relating to the vaccine platform, the immunization regimen, the testing subjects, and the trial size[2]. A small clinical trial may not provide sufficient information and statistical power to determine the efficacy and to inform on the effective use of the vaccine in humans[32]. However, technologies that enable in-depth analysis of vaccine-induced response may prove valuable for vaccine evaluation at this very early stage. NGS provides a powerful, quantitative tool to probe the vaccine-induced antibody response in humans[8–10,13]. However, the potential of NGS-based vaccine evaluation has not been fully explored, as previous studies were mainly focused on the global repertoires or vaccine-specific B-cell populations[8–10]. Recently, the influenza vaccine-induced bNAb lineages were analyzed that placed a premium on the structural recognition of hemagglutinin (HA) rather than the longitudinal bNAb lineage development[11]. In our previous study,

we have characterized the polyclonal serum response to Hecolin® in humans[18,22]. Here, we combined antigen-specific single-cell sorting, repertoire NGS, and antibody lineage tracing to deconvolute the polyclonal serum response to achieve a better understanding of HEV vaccine-induced antibody response in humans.

The NGS-derived antibody-repertoire profiles revealed the complexity of the human B-cell response to three vaccine doses over a period of 7 months. The steady signals observed in conventional serological assays were constituted by rapid responses from antibody families of diverse genetic makeups overlapping in time. Nonetheless, common patterns were identified for germline gene frequency, timing of the plasmablast response, vaccine-stimulated affinity maturation, and preferred CDRH3 length, which were consistent with serum and mAb analyses. Lineage traceability appeared to an indicator of antibody function. Most of the p239(1)-specific mAbs could be found after the second dose, suggesting that the booster doses are crucial for Hecolin®, a recombinant vaccine that requires multiple doses to stimulate the immune system[2,33]. Lineage expansion was most visible after the last dose given 6 months after the first dose, consistent with the previous studies of the benefit of long boost intervals[2,33]. This also supports the finding that participants who received two doses of Hecolin® at months 0 and 6 showed higher geometric mean antibody titers (GMTs) than those who received two doses at months 0 and 1[22]. For vaccines against pertussis, hepatitis A and B, prolonged intervals are correlated with increased antibody responses, including GMTs and seroprotection rates[34–36]. Therefore, the three-dose schedule (months 0, 1, and 6) may be crucial for the development of long-term protective immunity of a recombinant vaccine such as Hecolin®[22].

The dynamics of anti-HEV IgG in four donors showed typical primary and secondary immune responses, in which anti-HEV IgG was detectable at 1 month after the first dose and showed a notable increase at 1 week after the second and the third dose. With the memory B cells generated following the primary response, the secondary response occurred earlier and significantly stronger[37]. Nearly 50% of the traceable vaccine-specific antibody lineages peaked at 1 month after the first dose, and almost all peaked at 1 week after the second and the third dose. The vaccine-specific antibody lineages declined to the baseline within a month after the second and the third dose, which suggests the formation of B-cell memory following the plasmablast responses to the vaccine boosts[24]. However, the anti-HEV IgG in serum showed a continuous increase, which could be explained by the accumulation of secreted antibodies with higher affinity, or by the expanded PC population in the secondary lymph nodes, spleen, and bone marrow (a major site for long-lived PCs)[38].

The efficacy of Hecolin[14,22] appeared to be achieved with diverse antibody responses. Donor 1 presents an example of ideal vaccine response, as the p239(1)-specific mAbs from this donor showed substantial antigen-binding and neutralizing activity with a large population of plasmablasts in the repertoire. In contrast, the p239(1)-specific mAbs from donor 4 showed the lowest level

of antigen binding and neutralization with a small population of plasmablasts in the repertoire, consistent with the low serum neutralization, anti-HEV IgG titer, and anti-HEV avidity. Notably, donors 2 and 3 exhibited similar immune outcomes (e.g., anti-HEV IgG titers) with very different responses to vaccination: the p239(1)-specific mAbs from donor 2 showed lower antigen binding and neutralizing activity than those from donor 3, but with a larger plasmablast population in the repertoire. These results indicated that the protective antibody response against HEV can be generated via different mechanisms. Compared with donor 1, a higher frequency was observed for *IGHV3-48* and *IGHV3-53* in donors 2, 3, and 4. However, no HEV p239(1)-specific mAbs from these three donors were of the *IGHV3-48* and *IGHV3-53* origin, suggesting that the antibodies arisen from these two germline genes were not directed to the vaccine antigen. Of note, two p239(1)-specific mAbs from donors 2 and 4 were found in the Pre repertoires with dramatic lineage expansion during the vaccination, suggesting that the immune history may skew the antibody response away from vaccination[39]. Antibody function is determined by both the variable and the constant regions[40]. Since our functional assays were focused on antigen binding and virus neutralization, which are encoded by the variable region, all HEV-specific mAbs in this study were expressed as full-length IgG1 irrespective of their original subtypes. The role of other antibody isotypes, as well as different IgG subtypes, in HEV vaccine-mediated protection may warrant further investigation in future studies.

In summary, our study provides critical insights into the mechanism of cross-genotype protection conferred by the HEV vaccine, Hecolin®, and revealed the importance of the three-dose regimen. In a recent study, the NGS analysis of antigen-specific B-cell population revealed significant differences in the murine B-cell response to hepatitis C virus (HCV) vaccines based on the E2 glycoprotein core and an E2 core-presenting nanoparticle[41]. It is foreseeable that the antigen-specific B-cell sorting, at both single-cell and population levels, can be combined with repertoire NGS to investigate the mode of vaccine action and to quantitatively assess vaccine-induced antibody response during immunization. Such an integrated analytical approach will prove useful for future vaccine evaluation and comparison in a clinical setting.

## Methods

**Participants and vaccination.** Four healthy adult donors (aged 19–25 years) with no prior history of HEV vaccination or infection were recruited with informed consent. These donors were both anti-HEV antibody (including anti-HEV IgM and IgG) negative and HEV pathogen (HEV RNA and HEV antigen) negative. All donors were given three doses of the HEV vaccine, Hecolin® (Xiamen Innovax, Xiamen, China), according to a standard schedule (0, 1, and 6 months) and followed-up for 376 days after the first dose (Supplementary Table 1; Fig. 1a). Sera and PBMCs were collected before vaccination, 1 week and 1 month following each dose, and at additional time points (Fig. 1a). This study was designed in accordance with the Declaration of Helsinki and subsequently approved by the medical ethics committee of the School of Public Health, Xiamen University.

**Serology testing.** HEV antigen in the donor sera was tested using the commercial ELISA kits with mAbs 12F12 and #4[42], according to the manufacturer's instructions (Wantai, Beijing, China). HEV RNA was purified from a volume of 50 μL of sera and tested by the real-time reverse transcriptase PCR (RT-PCR) using a CFX96 real-time system and C1000 thermal cycler device (Bio-Rad, Hercules, CA) with the primer pair of JVHEVF: 5′-GGTGGTTTCTGGGGTGAC-3′ and JVHEVR: 5′-AGGGGTTGGTTGGATGAA-3′ and the probe of JVHEVP: 5′-TGATTCTCAGCCCTTCGC-3′[42]. Anti-HEV IgM and IgG in the donor sera were detected using the commercial ELISA kits (Wantai, Beijing, China). The concentration of anti-HEV IgG was measured against a reference WHO serum[14]. The sera were also tested for anti-HEV IgG avidity using the modified anti-HEV IgG assay[22]. Briefly, sera were tested for anti-HEV IgG in the presence and absence of 5 M urea. Paired serum samples were tested for anti-HEV IgG quantification. While one serum sample was tested according to the manufacturer's instructions, testing of the other sample was modified by replacing the wash buffer with a 5 M urea in wash buffer. Anti-HEV IgG avidity was the residual antibody level in the presence of urea relative to that in the absence of urea. The initial immune response to viral

infection is often characterized by the generation of low-avidity antibodies[43], which will eventually be replaced by antibodies with higher avidity at the convalescence stage[44].

**Expressions and labeling of proteins.** The recombinant HEV capsid proteins p239, E2 dimer, E2a dimer were cloned into a pTO-T7 expression plasmid, which were subsequently expressed in the *E. Coli* ER2566 strain as previously reported[20,21,45,46]. Mutants were generated by site-directed mutagenesis using polymerase chain reaction (PCR) and expressed in the *E. Coli* ER2566 strain following the same procedure as for the wild-type proteins[18]. Briefly, the recombinant proteins form inclusion bodies in host cells. The inclusion bodies were treated with 2% Triton X-100 at 37 °C for 30 min and dissolved in 4 M urea. The supernatant was dialyzed against PBS (pH 7.4). Biotin labeling of p239(1) was performed according to the manufacturer's instructions (Pierce).

**Flow cytometry and single B-cell sorting.** PBMCs from four donors were stained by a cocktail of live per dead Aqua Dead Cell Stain Kit (Invitrogen, Molecular Probes, L34957, dilution: 1 per 100), CD3-PE-Cy7 (BD Biosciences, 557851, dilution: 1 per 200), CD19-BV786 (BD Biosciences, 563325, dilution: 1 per 200), CD27-BV650 (BD Biosciences, 563228, dilution: 1 per 100), CD20-FITC (BD Biosciences, MHCD2001, dilution: 1 per 100), CD38-PE (BD Biosciences, 555460, dilution: 1 per 100), anti-human IgG-BV421 (BD Biosciences, 562581, dilution: 1 per 100), and biotinylated p239(1) for 30 min on ice, followed by Streptavidin–allophycocyanin (SA-APC) for additional 30 min on ice. Plasmablasts were identified as CD3−/CD20−/CD19+/CD27+/CD38+ cells as previously reported[24,47] and shown in Supplementary Fig. 2, and IgG+ memory B cells were identified as CD3−/CD20+/CD27+/IgG+ cells and shown in Supplementary Fig. 2. IgG+ memory B cells that bind to p239(1) were single-cell sorted from donor samples collected at 3M1 from each donor. Single cells were sorted by fluorescence-activated cell sorting on an Aria III sorter (BD Biosciences) into 96-well PCR plates containing 20 μL per well of lysis buffer [5 μL of 5 × first strand buffer (Invitrogen), 1.25 μL dithiothreitol (Invitrogen), 0.5 μL RNase Out (Invitrogen), 0.0625 μL Igepal (Sigma)]. Plates were stored at −80 °C prior to reverse transcription of RNA.

**Antibody expression.** Antibody variable genes (IgH, Igλ, and Igκ) were amplified by RT-PCR and nested PCR reactions with primers listed in Supplementary Table 8[48]. The paired heavy and light chains were then cloned into the full-length IgG1 heavy- and light-chain vectors for mAb expression[48]. Here, the mAb name is defined as the donor index (A, B, C, and D for donors 1, 2, 3, and 4, respectively) with a uniquely assigned clone number, e.g., A103. Heavy-chain sequences identified from the NGS-derived donor repertoires were synthesized, cloned into the IgG1 heavy-chain expression vector, and paired with their respective parental light chains. For heavy-chain variants, the nomenclature is defined as [Parental mAb]_[Time point]_[Sequence number]. Full-length IgGs were then transiently expressed in ExpiCHO cells with equal amounts of heavy and light-chain plasmids according to the manufacturer's instructions (Life Technologies) and purified using a recombinant protein-A column (GE Healthcare).

**Sample preparation using 5′-RACE PCR.** Sample preparation was done using 5′-rapid amplification of cDNA ends (RACE) PCR with the primers listed in Supplementary Table 9 and reported in our previous studies[6,7]. Briefly, the total RNA was purified from 5 to 10 million PBMCs into 30 μL of water using RNeasy Mini Kit (Qiagen, Valencia, CA). For the unbiased repertoire analysis, 5′-RACE was performed using SMARTer-RACE cDNA Amplification Kit (Clontech). The cDNA was purified and eluted in 20 μL of elution buffer (NucleoSpin PCR Clean-up Kit, Clontech). The immunoglobulin PCRs were conducted with Platinum Taq High-Fidelity DNA Polymerase (Life Technologies) in a volume of 50 μL, containing 5 μL of cDNA as a template, 1 μL of 5′-RACE primer, and 1 μL of 10 μM reverse primer. To facilitate NGS on the Ion GeneStudio S5 system or the Ion Personal Genome Machine (PGM) system, the forward 5′-RACE primer contained a P1 adaptor, while the reverse primer contained an A adaptor and an Ion Xpress™ barcode (Life Technologies) to differentiate the antibody libraries from different time points. A total of 25 cycles of PCRs were performed and the PCR products (~600 bp) were gel purified (Qiagen)[6,7].

**Next-generation sequencing of antibody-repertoire libraries.** Antibody NGS has been adapted to the Ion GeneStudio S5 system[31,49]. Briefly, the antibody heavy-chain libraries were quantitated using Qubit® 2.0 Fluorometer with Qubit® dsDNA HS Assay Kits. Equal amounts of the heavy-chain libraries from seven time points were mixed and loaded onto an Ion 530 chip to increase the sequencing depth and to eliminate run-to-run variation. Template preparation and (Ion 530) chip loading was performed on the Ion Chef system using Ion 530 Ext Kits, followed by S5 sequencing with the default settings. An additional NGS experiment was performed to increase the repertoire coverage for donor 4. Specifically, the mixed Pre, 1M1, 2M1, and 3M1 libraries were sequenced on the Ion PGM system using a 318 v2 chip with PGM™ Hi-Q 400 Kit for a total of 850 nucleotide flows[6,7]. The datasets obtained from PGM were combined with those from S5 prior to data processing and bioinformatics analysis. Raw data were processed without 3′-end trimming in

base calling to extend the read length. The human *Antibodyomics* pipeline version 1.0 has been modified to improve data accuracy and computational efficiency[49]. This pipeline was used to process and annotate the antibody NGS data of four HEV-vaccinated donors for repertoire profiling and lineage tracing[6,7]. The distributions of germline gene usage, germline divergence or degree of SHM, and CDR3 loop length were derived from the NGS data as antibody-repertoire profiles. Two-dimensional (2D) divergence/identity plots were used to visualize HEV p239 (1)-specific antibody lineages in the context of total heavy-chain repertoire. A CDRH3 identity of 90% was used as the cutoff for identifying sequences evolutionarily related to a reference antibody (shown as magenta dots on the 2D plots). The CDRH3-defined somatic variants were further divided into groups based on an overall identity cutoff of 90%. The sequence at the center of each group, or cluster, was used as the representative for antibody synthesis and functional characterization.

**Antibody germline gene usage and IgG subtype identification**. Germline gene usage was analyzed for the variable region of IgG heavy and light chains using the IMGT V-quest webserver[50]. IgG subtype was identified based on the specific sequence motif in the CH1 region as previously described[40]. Briefly, IgG1, IgG3 and IgG2/4 were identified by the sequences of 5′-AAGAGCACCTCT-3′, 5′-AGGGAGCACCTCT-3′, and 5′-AGGGAGCACCTCC-3′, respectively. Subsequently, IgG2 and IgG4 were separated from each other by sequences of 5′-GCCTCCACC AAGGGC-3′ and 5′- GCTTCCACCAAGGGC-3′, respectively.

**Neutralization assays**. HEV neutralization was performed on HepG2/C3A cells using a previously described protocol with some modifications[51]. One day before infection, HepG2/C3A cells ($3 \times 10^4$) were seeded onto 96-well plates. Sixfold serially diluted mAbs or fourfold serially diluted sera were first incubated with purified virus at 37 °C for 30 min before inoculation of the HepG2/C3A cells. The cell supernatant was tested for HEV antigen 5 days after inoculation. Neutralization curves were fit using a nonlinear regression model (GraphPad Software Inc., San Diego. CA) to calculate the $IC_{50}$. The highest concentration will be designated as the $IC_{50}$ of an mAb or serum if the first dilution displayed less than or equal to 50% neutralizing activity.

**Indirect ELISA**. The binding activity of the mAbs were determined using an indirect ELISA as previously described[18]. Briefly, the mAbs were added to antigen-coated (E2, E2a, or p239, 100 ng per well) microwell plates and incubated at 37 °C for 30 min. After five washes, HRP-conjugated anti-human IgG antibodies were added to detect the bound mAbs. After incubation at 37 °C for 30 min, the plates were washed five times and 100 μL of tetramethylbenzidine substrate solution was added to the wells. The reaction was stopped by adding 50 μL of 2 M $H_2SO_4$ after incubation at 37 °C for 15 min. The optical density (OD) was measured at 450 nm with a reference wavelength of 620 nm.

**Western blot**. The western blot was performed according to the previously described methods except that the HRP-conjugated anti-human IgG antibodies were used instead of the anti-murine antibodies[18]. Briefly, the recombinant p239(1) proteins with or without boiling were loaded onto SDS-PAGE gel and subsequently electroblotted onto nitrocellulose membrane at 35 mA for 55 min. The membranes were blocked with 5% skim milk and then incubated with sera or p239(1)-specific mAbs for 30 min. After rinse, the membranes were incubated with anti-human HRP-conjugated antibody (Sino Biological inc, Cat# SSA001). Color was developed over a period of 5 min using SuperSignal West Femto Maximum Sensitivity Substrate (PIERCE).

**Competitive ELISA (cELISA) and cluster analysis by SPSS**. For vaccine-induced mAbs, epitope specificity was determined by cELISA using a panel of conformation-dependent murine mAbs that recognize six different antigenic clusters (C1–C6) on the HEV capsid as previously reported[18]. Briefly, the unlabeled human mAbs (50 μg per well) or PBS were added to p239-coated 96-well microplates and then incubated for 30 min at 37 °C. Next, HRP-conjugated murine mAbs were added at selected dilutions that resulted in OD readings of ~1.5 in the indirect ELISA. After incubation for 30 min at 37 °C, the microplates were rinsed, and the color was developed. The blocking efficiency was measured quantitatively by comparing OD in the presence or absence of competitor mAbs. The cELISA data were processed and transformed using the formula $\log_2 (OD_{inhibited}/OD_{original})$ and clustered with SPSS 17 (SPSS Inc., Chicago, USA) using the "between-group linkage" method and the "cosine" interval method as previously reported[18].

**Statistical analysis**. Statistical significance (e.g., in Fig. 2f and Supplementary Fig. 8) was determined using the Spearman rank-correlation analysis. Vaccine-specific mAbs that failed to neutralize HEV were excluded from the statistical analysis because their $IC_{50}$ values cannot be accurately determined. *P* values were calculated from the two-tailed test and only *P* values of 0.05 or lower would be considered statistically significant.

**Reporting summary**. Further information on research design is available in the Nature Research Reporting Summary linked to this article.

## Data availability

All the repertoire NGS data sets can be obtained from the NIH Sequence Read Archive (SRA) with the identifier SRP265670. The sequences of p239(1)-specific mAbs isolated from four donors, as well as the heavy-chain variants derived from the NGS-based lineage analysis, are summarized in Supplementary tables. Source data are provided with this paper.

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

## Acknowledgements

This work was supported by the National Natural Science Foundation of China (Grant Nos. 81871247, 31730029, and 81701576), the Major Infectious Disease Project of China (Grant No. 2018ZX10101001-002), and the Scientific Research Foundation of State Key Laboratory of Molecular Vaccinology and Molecular Diagnostics (Grant No. 2018ZY001). This work was funded in part by HIV Vaccine Research and Design (HIVRAD) program (P01 AI124337) (to J. Zhu), NIH Grants R01 AI129698 and R01 AI140844 (to J. Zhu).

## Author contributions

Study design by G.-P.W., L.H., Z-.Z., J. Zhang, J. Zhu, and N.-S.X.; donor sample collection and characterization by G.-P.W., S.-L.W., C.L., and S.-J.H.; antibody isolation by G.-P.W., Z.-M.T., S.-L.W., Y.-Z.C., and W.-X.L.; synthesis of antibody genes and construction of expression plasmids by S.-L.W., X.Z., Z.-H.C., and Y.-B.W.; expression and purification of antibodies and HEV proteins by G.-P.W., S.-L.W., X.Z., J.-X.C., and S.-W.L.; in vitro characterization of antibodies by G.-P.W., Z.-M.T., S.-L.W., X.Z., J.-X.C., C.L., and D.Y.; library preparation and repertoire NGS by L.H. and J. Zhu; antibodyomics and lineage analyses by X.L., L.H., and J. Zhu; paper written by G.-P.W., Z.-Z.Z., and J. Zhu.

## Competing interests

The authors declare no competing interests.
