## [Peer Review File · Nature Communications]

Reviewers' Comments:

Reviewer #1:

Remarks to the Author:

The study examined the antibody immune response of four subjects that were immunized with the hepatitis E virus vaccine, Hecolin. After vaccination, the IgG response was detected in the 4 subjects. The amount and avidity of the antibodies increased during the period of the experiment. Serum antibodies showed an increase in neutralization capacity during the experiment. Single-cell sequencing of the immunoglobulins revealed the predominant V regions of the response and low SHM levels. The antibodies show medium-strong antigen-binding activity, neutralizing activity and binding to the E2 domain of p239 through binding of C2, C5, and C6 antigenic sites. NGS sequencing revealed the V region distribution of the antibodies, and that increase SHM takes place only after the third boost. The authors analyzed antibody clonal members that can be traced based on CDR3 similarity. The number of traceable antibodies increased with each boost. Traceable antibodies that were detected after the first dose were found throughout the experiment. Persisting clones that were found after all the boosts showed increase binding capacity. Analysis of dendrograms and antigen-binding analyses suggest that the source of the high-affinity antibodies are from memory recall response. Based on these parameters it seems like some of the donors responded more effectively than others.

This study is very well done, clearly written and of high importance. I did not find any flow or problems with the study. The design of the experiments and analysis were very clear and easy to understand. The conclusions are supported by the results. The similar and different response patterns of the different subjects were interesting and important.

Reviewer #2:

Remarks to the Author:

Wen and colleagues performed a comprehensive study using next generation sequencing to study the antibody response to HEV vaccination coupled with basic molecular biology to validate a portion of the big data. The body of work is impressive and is an approach that can be emulated in future studies. However, there are several issues that should be addressed before publication.

1. Overall, there might be some confusion to what the authors are studying concerning the immediate antibody response following days and weeks after vaccination and the memory B-cell response they characterized following the third immunization. Please notes below and try to clarify.
2. Based off of the cell surface markers in Figure S2 (CD27 and CD38) that authors used to gate and isolate the monoclonal antibodies, it appears that these cells are plasmablasts and not plasma cells.
3. Authors should discuss the significance and significance of anti-HEV antibodies to the p239(1) dimer. Do dimers represent a better epitope for protection/neutralization? Is it expressed on the surface of infected cells, etc. Authors should further explain why differential binding properties of the antibodies.
4. Authors should discuss the significance and implications of IgG avidity of anti-HEV antibodies. It is not clear why avidity of the antibody response is being measured; the methods section describing this assay should also be expanded. From reading the text, people who are not too familiar with the virus or antibody biology, will not understand why avidity may be important in HEV immunity? Is avidity typically investigated in other systems, for example. Please clarify.
5. It may help readers if the authors could introduce the E2 domain in the introduction – its role in the virus replication cycle and its sequence conservation between the 4 genotypes.
6. It is a bit confusing when the authors discuss memory B-cell since according to their gating strategies, they are still using the plasmablast gating strategy to characterize the memory B-cell. Authors should clarify.
7. In Figure 4, donor 1, who had responded the best to the vaccination regimen had very little or no antibodies in the VH3-48 and 3-53 compared to donors 2, 3 and 4. Can the authors discuss this and perhaps highlight what these antibodies target in donors 2, 3 and 4?
8. What is surprisingly not discussed and reported are the isotypes of the original antibodies that were validated. This should be reported and discussed. We know from past studies that isotypes can play a major role in Fc-dependent cell-mediated mechanisms.

Point-by-point response to the reviewers' comments

Reviewer #1 (Remarks to the Author):

General comments: *The study examined the antibody immune response of four subjects that were immunized with the hepatitis E virus vaccine., Hecolin. After vaccination, the IgG response was detected in the 4 subjects. The amount and avidity of the antibodies increased during the period of the experiment. Serum antibodies showed an increase in neutralization capacity during the experiment. Single-cell sequencing of the immunoglobulins revealed the predominant V regions of the response and low SHM levels. The antibodies show medium-strong antigen-binding activity, neutralizing activity and binding to the E2 domain of p239 through binding of C2, C5, and C6 antigenic sites. NGS sequencing revealed the V region distribution of the antibodies, and that increase SHM takes place only after the third boost. The authors analyzed antibody clonal members that can be traced based on CDR3 similarity. The number of traceable antibodies increased with each boost. Traceable antibodies that were detected after the first dose were found throughout the experiment. Persisting clones that were found after all the boosts showed increase binding capacity. Analysis of dendrograms and antigen-binding analyses suggest that the source of the high-affinity antibodies are from memory recall response. Based on these parameters it seems like some of the donors responded more effectively than others.*

This study is very well done, clearly written and of high importance. I did not find any flow or problems with the study. The design of the experiments and analysis were very clear and easy to understand. The conclusions are supported by the results. The similar and different response patterns of the different subjects were interesting and important.

Response:

We thank the reviewer for the careful reading of our manuscript, and gratefully appreciate the positive comments on the scientific rigor and importance of our study. In the revised manuscript, we have included additional data to address the reviewers' comments and have improved the flow and clarity of the writing.

Reviewer #2 (Remarks to the Author):

General comments: *Wen and colleagues performed a comprehensive study using next generation sequencing to study the antibody response to HEV vaccination coupled with basic molecular biology to validate a portion of the big data. The body of work is impressive and is an approach that can be emulated in future studies. However, there are several issues that should be addressed before publication.*

Response:

We thank the reviewer for the positive comments and for noting the implication of our findings for future vaccine studies. We have addressed all the reviewer's comments and revised the manuscript accordingly. All changes in the manuscript are highlighted in yellow.

1. Overall, there might be some confusion to what the authors are studying concerning the immediate antibody response following days and weeks after vaccination and the memory B-cell response they characterized following the third immunization. Please see notes below and try to clarify.

Response:

We thank the reviewer for noting this potential confusion. Based on the reviewer's comments, we have revised the manuscript on pages 5, 6, 12, 17, 18, and 21.

2. Based off of the cell surface markers in Figure S2 (CD27 and CD38) that authors used to gate and isolate the monoclonal antibodies, it appears that these cells are plasmablasts and not plasma cells.

Response:

We have modified Fig. S2 to indicate the cell populations used for plasmablast analysis and antibody isolation and included a more detailed description of surface markers in the figure legend. We have also revised the manuscript, on page 6, to define the surface markers used in cell sorting and modified Fig. S3 to change “plasma cells” to “plasmablasts”.

Briefly, in this study, CD27+/CD38+ cells were identified as plasmablasts according to a previously described protocol (Smith et al., *Nat Protoc* 4:372-384, 2009). Although the HEV vaccine antigen was not used in the sorting experiment, the plasmablasts were most likely specific to the vaccine antigen as the PBMCs were collected during the vaccination (Smith et al., *Nat Protoc* 4:372-384, 2009). Based on the reviewer’s comment, we have revised the manuscript to change “plasma cells” to “plasmablasts” on pages 5, 6, 12, 17, 18, and 21. IgG+ memory B cells (defined as CD3-/CD20+/CD27+/IgG+) that bind to p239(1) – not plasmablasts or plasma cells – were single-cell sorted for the isolation of monoclonal antibodies (mAbs) to facilitate longitudinal repertoire analysis and antibody lineage tracing. This point has now been clarified in the revised manuscript.

3. Authors should discuss the significance and significance of anti-HEV antibodies to the p239(1) dimer. Do dimers represent a better epitope for protection/neutralization? Is it expressed on the surface of infected cells, etc. Authors should further explain why differential binding properties of the antibodies.

Response:

We thank the reviewer for this comment. HEV is a positive-sense, single-stranded, nonenveloped, RNA icosahedral virus. The cryo-EM structure of HEV VLP (PDB ID: 6LAT) shows an array of dimeric E2s domains on the particle surface (Zheng et al., *PNAS U S A*, 116, 26933-26940, 2019). It has been reported in multiple studies that dimerization of E2s domain is essential for virus-host interaction and cross-genotype NAb bind to conformational epitopes on the dimeric interface (Li et al., *PLOS Pathog*, 5(8):e1000537, 2009; Gu et al., *Cell Res* 25:604-620, 2015; Zhao et al., *J Bio Chem*, 290(32):19910-22, 2015). Based on the reviewer’s suggestion, we have revised the manuscript on page 9.

4. Authors should discuss the significance and implications of IgG avidity of anti-HEV antibodies. It is not clear why avidity of the antibody response is being measured; the methods section describing this assay should also be expanded. From reading the text, people who are not too familiar with the virus or antibody biology, will not understand why avidity may be important in HEV immunity? Is avidity typically investigated in other systems, for example. Please clarify.

Response:

We thank the reviewer for noting this important point. The initial immune response to viral infection is characterized by the generation of low-avidity antibodies, which will eventually be replaced with high-avidity antibodies during the convalescence stage. Thus, as the infection proceeds, IgG avidity gradually increases over time. A similar pattern of increasing IgG avidity can be observed in vaccination due to repeated antigen exposure in a multidose regimen. Based on the reviewer’s comment, we have modified the manuscript to discuss the significance and implications of IgG avidity on pages 5 and 20.

5. *It may help readers if the authors could introduce the E2 domain in the introduction – its role in the virus replication cycle and its sequence conservation between the 4 genotypes.*

Response:

In the revised manuscript, we have included the description of the E2s domain on page 4.

6. *It is a bit confusing when the authors discuss memory B-cell since according to their gating strategies, they are still using the plasmablast gating strategy to characterize the memory B-cell. Authors should clarify.*

Response:

We have addressed this confusion in our response to the reviewer's question #2. Briefly, more details about the gating strategy used to sort memory B cells have been included in a new version of Fig. S2 and in the figure legend. We have also revised the manuscript accordingly, on page 6.

7. *In Figure 4, donor 1, who had responded the best to the vaccination regimen had very little or no antibodies in the VH3-48 and 3-53 compared to donors 2, 3 and 4. Can the authors discuss this and perhaps highlight what these antibodies target in donors 2, 3 and 4?*

Response:

We thank reviewer for the comment. Compared with donor 1, a higher frequency of germline usage was observed for IGHV3-48 and IGHV3-53 in donors 2, 3 and 4. However, no HEV p239(1)-specific mAbs isolated from these three donors were of the IGHV3-48 and IGHV3-53 origin. The target of the antibodies corresponding to the increased IGHV3-48 and IGHV3-53 usage were unknown and will be an important topic to investigate in our follow-up studies. Nonetheless, we have revised the manuscript to report this finding on page 18.

8. *What is surprisingly not discussed and reported are the isotypes of the original antibodies that were validated. This should be reported and discussed. We know from past studies that isotypes can play a major role in Fc-dependent cell-mediated mechanisms.*

Response:

We agree with the reviewer that the isotypes of the sorted antibodies may be of importance. However, this study was mainly focused on the *in vitro* antibody function such as antigen binding and neutralizing activity, which are encoded by the variable region and are independent of the isotype determined by the constant region. Nonetheless, following the reviewer's suggestion, we have included a new figure in Supplemental Material (Fig. S5e) to summarize the isotypes of HEV p239(1)-specific mAbs. We have also revised the manuscript to report our findings on pages 7 and 18.